

# Climatic niche comparison across a cryptic species complex

Qing Zhao[1,*], Hufang Zhang[2,*] and Jiufeng Wei[1]

[1] Department of Entomology, Shanxi Agricultural University, Taigu, Shanxi, P. R. China
[2] Department of Biology, Xinzhou Teachers University, Xinzhou, Shanxi, P. R. China
[*] These authors contributed equally to this work.

## ABSTRACT

According to current molecular evidence, the *Chionaspis pinifoliae heterophyllae* species complex has been recognized as 10 cryptic species. In this study, we construct potential distribution maps for seven cryptic species based on climatic variables. This was done to assess the main environmental factors that have contributed to the distribution map and test the degree of niche overlap across the seven cryptic species. We used MaxEnt to build the climatic niche models under climatic variables. For these models, the similarities and differences of the niches across the cryptic species were estimated. By comparing the potential distribution model of each cryptic species, our results suggested parapatric, sympatric and allopatry populations for this cryptic species complex. Our results showed high variability in niche overlap, and more often niche conservatism than niche divergence. The current species delimitation of the *Chionaspis pinifoliae heterophyllae* complex by molecular information and the hypothesis that the niche overlap in the sympatric population is higher than that of the allopatry population were supported based on the findings. This study will provide baseline data and a distribution range to facilitate the further control of these insects and formulate quarantine measures.

# INTRODUCTION

Ecological factors play an important role in speciation events by providing sources of selection that drive micro-evolutionary change and by imposing constraints that limit organism performance (*Barrett & Hoekstra, 2011*; *Pyron & Burbrink, 2009*; *Rissler & Apodaca, 2007*). If evolutionary lineages become geographically isolated during periods of climatic change, ecological components can also directly drive speciation. Nevertheless, species can also maintain their ancestral niche, which is referred to as niche conservatism (*Wiens, 2004*; *Peterson, Soberón & Sanchez-Cordero, 1999*; *Peterson, 2011*). The study of how species fluctuate in their requirements for and tolerance of such factors has advanced, in part, due to the continued theoretical development and quantification of the ecological niche of species (*Soberón, 2007*). Many methods have been proposed to test niche conservatism (*Wiens, 2004*) and define the climatic variables that constrain the distribution of species. Ecological niche modeling (ENM) is a relatively new method that has been used as a powerful tool to examine niche divergence and conservatism (*Culumber*

Corresponding author
Jiufeng Wei, wjfeng@nwsuaf.edu.cn

*& Tobler, 2016*; *Scriven et al., 2016*). Spatially unequivocal environmental data and models allow for extensive scale tests about whether speciation is associated with niche divergence or whether closely related species tend towards niche conservatism (*McCormack, Zellmer & Knowles, 2010*). The development of ENM has promoted the extraction of ecological niche characteristics, which can help identity the niche limits of cryptic species (*Wellenreuther, Larson & Svensson, 2012*). Niche features have been applied to improve inferences about species boundaries and speciation mechanisms. Recently, a principal component analysis (PCA) method was present by *Broennimann et al. (2012)* that transforms the investigated environmental variables into a two-dimensional space identited by the first and second principal components. The niche overlaps value was measured by this approach that provides a relatively reliable way to test the niche divergence and niche conservatism hypotheses (*Ahmadzadeh et al., 2013*).

Cryptic species are a distinct but morphologically similar species that were classified as being a single one (*Pfenninger & Schwenk, 2007*). These species are not only important for taxonomic reasons, but also hold significant meaning in terms of biogeography (*Pfenninger & Schwenk, 2007*) and biodiversity (*Oliver et al., 2009*). The armored scale insects (Hemiptera, Diaspididae) are a family of over 2,650 described species according to current records (*García et al., 2016*). The *Chionaspis pinifoliae heterophyllae* species complex has only included two species, which are *C. heterophyllae* Cooley and *C. pinifoliae* (Fitch), since 1921 (*MacGillivray, 1921*). However, current reanalysis of species diversity within this group by *Gwiazdowski et al. (2011)* that suggested the presence of at least 10 closely related species, which have been delimited as cryptic species. These species are origin to North America (*Vea, Gwiazdowski & Normark, 2012*) and are considered pests on *Pinus* in forests and ornamental settings (*Miller & Davidson, 2005*). The adaptive deme formation (ADF) hypothesis proposes that a population may adapt to an individual host resulting in "a series of semi-isolated subpopulations, or demes" (*Edmunds & Alstad, 1978*). Many studies support ADF leading to multiple closely related cryptic species, such as armored scales (*Anderson et al., 2010*; *Cook & Rowell, 2007*). The result of research of *Gwiazdowski et al. (2011)* also supports this hypothesis. However, this study did not consider the ecological or climatic requirements of the species. Identifying the ecological niches of cryptic species is important to understanding the creation and maintenance of biodiversity (*Oliver et al., 2009*).

In the current study, ENMs and ordination techniques were used to characterize the ecological niches of the members of the *Chionaspis pinifoliae heterophyllae* species complex and test similarities between them. The main questions addressed in this study were: (1) What are the main environmental variables that constrain the potential distributions of members of *C. pinifoliae heterophyllae*? (2) Do cryptic species share the same ecological conditions? (3) What is the potential mechanism of geographic speciation of the allopatric and sympatric populations? The closely related species will result in high levels of similarity, ecological niche and spatial overlap. In addition, with niche divergence as a speciation mechanism, we would expect the ecological niches of closely related species to differ significantly. Because the cryptic species have similar morphological and physiological features, their ecological niches would be similar. However, because of the different

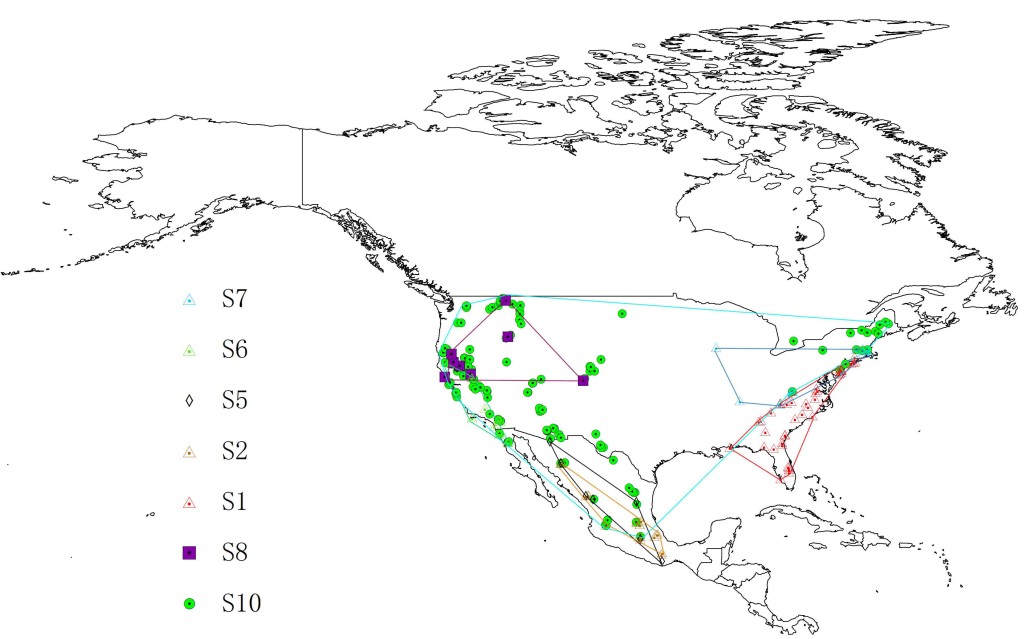

**Figure 1** **Study areas and distribution locations for seven cryptic species in geographical space.** The figure show a geographic minimum convex polygon. Different symbols represent different species. The base map was created with Natural Earth Dataset (http://www.naturalearthdata.com).

adaptations to different environmental conditions, the non-equivalence of their ecological niches would be expected in the current study.

## METHODS

### Study area and species

The *Chionaspis pinifoliae-Chionaspis heterophyllae* (CPCH) species complex inhabits across the USA, Mexico, and parts of Canada. Distribution data for CPCH were taken from *Gwiazdowski et al. (2011)*. S3, S4, and S9 could not be included because of the low number of available sample locations (<5). Duplicate occurrences with the same geographic coordinates were removed using ENMtools software (*Warren, Glor & Turelli, 2010*). Occurrence records are often biased towards areas that are easily accessible or are near cities or other areas of high population density, thus, to remove spatial autocorrelation and the sampling bias, a grid of 5 × 5 km cells was created. A single point from each cell containing one or more sampling points was randomly selected. After filtering, 201 localities were retained in the final analysis, including S1 (29 points), S2 (21 points), S5 (8 points), S6 (22 points), S7 (22 points), S8 (13 points) and S10 (160 points). The distribution of the localities used in the study is presented in Fig. 1.

### Climate variables

Climate data for 19 environmental variables were obtained from the WorldClim database (*Hijmans et al., 2005*; http://www.worldclim.org). Climate data represent the biologically relevant summaries of means and variation in precipitation and temperature
recorded between 1950 and 2000. All climatic variables had a spatial resolution of 30 arc seconds (approximately 1 km resolution at the equator). Since strong covariance among environmental variables can cause "overfitting" of the model and affect model accuracy in ENM (*Synes & Bsborne, 2011*; *Boria et al., 2014*), it is important to minimize such correlations using dimension-reduction techniques. Thus, a Pearson's correlation test based on all 19 climate variables was conducted for all species' presence points, and used to exclude highly correlated variables ($|r| > 0.80$) from our models (Table S1). Six predictor variables were used in subsequent processing, which were Mean Diurnal Range (Bio2), Mean Temperature of the Wettest Quarter (Bio8), Mean Temperature of the Driest Quarter (Bio9), Annual Precipitation (Bio12), the Precipitation of the Warmest Quarter (Bio18) and the Precipitation of the Coldest Quarter (Bio19).

## Ecological niche modeling (ENM)

Ecological niche modeling uses environmental variables and occurrence data to simulate suitable environmental conditions for focus species. Researchers have proposed the use of many different software packages to study invasion biology (*Waltari & Hickerson, 2012*; *Guo et al., 2013*), conservation biology (*Préau et al., 2018*), biodiversity research (*Thuiller et al., 2006*) and ecological niches (*Chetan, Praveen & Vasudeva, 2014*; *Reeves & Richards, 2011*). Of the species distribution model algorithm methods, Maxent (Maximum Entropy) has proved powerful when modeling rare data with narrow ranges and only with presence-only data (*Elith et al., 2011*; *Qin et al., 2017*). The MaxEnt version (version 3.3.3k) (*Phillips, Anderson & Schapire, 2006*) was used to construct SDMs for all species.

The default MaxEnt settings were suggested to produce overfitted models in a recent study (*Radosavljevic & Anderson, 2014*). Hence, in order to balance the model fit and predictive ability, the R package ENMeval was used to select the optimal combination of two important MaxEnt's parameters which are the value of the regularization multiplier and the combination of feature classes (*Warren & Seifert, 2011*; *Muscarella et al., 2014*). The "checkerboard2" approach was employed to calculate the standardized Akaike information criterion coefficient (AICc), and models with the lowest delta AICc score were selected to run the final MaxEnt models. Five different feature class combinations (1) L; (2) LQ; (3) LQH; (4) LQHP, (5) LOHPT (where $L$ = linear; $Q$ = quadratic; $H$ = hinge; $P$ = product and $T$ = threshold) were tested, meanwhile, the regularization multiplier was varied from 0.5 to 4 in increments of 0.5. The result are shown in Table S2 and Fig. S1. The logistic output was used in the model, which was a continuous map with an estimated probability of presence between 0 (lowest) to 1 (highest). The percentages contribution and permutation importance of the environmental variables were calculated, and jackknife procedures were executed to evaluate the relative importance variable in MaxEnt. The model was run with a convergence threshold of $10^{-5}$, maximum iterations of 5,000 and maximum number of background points of 10,000. The MaxEnt model was created based on the 10-fold replicates with cross-validation method. The remaining model values were set to default values (*Penado, Rebelo & Goulson, 2016*).

The area under the receiver operating characteristic (ROC) curve (AUC) is a widely used statistic for the measurement of MaxEnt performance (*Merow, Smith & Silander, 2013*; *Wei*

*et al., 2018*). However, this method has been criticized for its equal weighting of omission and commission errors (*Lobo, Jiménez-Valverde & Real, 2008*; *Peterson, Papes & Soberón, 2008*). In addition, this method can not provide information about the spatial distribution of model errors the total extent to which models are carried out highly influences the rate of well-predicted absences, and the AUC scores (*Lobo, Jiménez-Valverde & Real, 2008*). Thus, an alternative Partial ROC metric approach was employed to model evaluations (*Peterson, Papes & Soberón, 2008*). Partial ROC statistics were implemented using online the Niche Toolbox site (http://shiny.conabio.gob.mx:3838/nichetoolb2/) with 1,000 replicates and $E = 0.05$. To improve predictions made in this study, the predicted continuous suitability maps by MaxEnt were converted into suitable/unsuitable areas (binary habitat) using an applied threshold. Here, the lowest presence threshold (LPT = minimum training presence threshold of MaxENT) was selected for each species (*Pearson et al., 2007*; *Wisz et al., 2008*). This threshold can identify the minimum predicted area possible whilst maintaining zero omission error in the training dataset.

## Calculating niche overlap in *E*-space

An assessment of niche overlap was used to quantify the niches shared within the species complex was done through. Climate niche overlaps among species were estimated using the PCA-env method proposed by *Broennimann et al. (2012)*. Principal component analysis (PCA) was used to transform the environmental space of the investigated or selected environmental variables into a two-dimensional space defined by the first and second principal components (*Strubbe, Beauchard & Matthysen, 2015*). The two-dimensional environmental space was then projected onto a $100 \times 100$ PCA grid of cells bounded by the minimum and maximum PCA values in the background data. This method also corrects the potential sampling bias in occurrence records using a smooth kernel density function (*Broennimann et al., 2012*). The niche overlap within the species complex was measured by the mean of Schoener's $D$ directly from the ecological niche space (*Warren, Glor & Turelli, 2008*). The Schoener's $D$ is an index which varies from 0 (no overlap) to 1 (overlap). In addition, there are two common statistical tests considering hypotheses (niche equivalence test and niche similarity test) of niche divergence or conservatism.

A niche equivalence test was applied to estimate whether the ecological niches of the pairs of the species complex are significantly different from each other and whether the two niche spaces are interchangeable. The niche equivalence was determined by comparing the niche overlap values ($D$) of pairs of a species complex to a null distribution of 100 overlap values. If the niche overlap value fell outside the 95% of the null hypotheses, the equivalency of the two niches could be rejected. Niche similarity test compares the niche overlap of one range randomly distributed over its background, while keeping the other unchanged ($1 \rightarrow 2$), and then carries out the reciprocal comparison ($1 \leftarrow 2$). For the similarity test, a $p$ value >0.05 was considered to indicate that niches were no more similar than expected by chance. Niche similarity tests were used in the current study to estimate niche differentiation. The overlap value between two ENMs was either above the 95% confidence interval of the null hypothesis, supporting niche conservatism, or below the 95% confidence interval of the null hypothesis, supporting niche divergence.

**Table 1  The LPT threshold value of each model from MaxEnt.**

| Species | S1 | S2 | S5 | S6 | S7 | S8 | S10 |
|---------|------|-------|-------|-------|-------|-------|-------|
| Value | 0.1779 | 0.509 | 0.586 | 0.580 | 0.305 | 0.540 | 0.029 |

**Notes.**
Average LPT (Lowest presence threshold), minimum training presence threshold.

In addition, the null hypothesis of niche equivalency was also tested using ENMTools 1.4.4 (*Warren, Glor & Turelli, 2010*). ENMTools uses MaxEnt to generate an ENM from each species, and then uses the model and predicted suitability scores generated by Maxent for each species to calculate niche equivalency test. This test is based on the metrics of niche overlap (Schoener's *D* and *I*) that ranging vary from 0 (no overlap) to 1 (complete overlap) (*Warren, Glor & Turelli, 2008*). The ecological interpretation of Schoener's *D* assumes that the suitability scores are proportional to species abundance, whereas Schoener's *I* treats the two ENMs as probability (*Warren, Glor & Turelli, 2010*). The null hypothesis that the two species pairs have similar niches is accepted if the niche overlap of both species pairs is outside of the 95% confidence interval. When niche overlap between two groups does not fall within the 95% confidence interval and Schoener's *D* and *I* are less than our assumption value, then the groups are separated (*Warren, Glor & Turelli, 2008*).

All GIS analyses were performed using ArcGIS version 10.2.1 (ESRI). All statistical analyses were performed in R using scripts from *Broennimann et al. (2012)*, now available in the 'ecospat' R package (*Di Cola et al., 2017*).

## RESULTS

### Niche modeling and responses to climate variables

The partial ROC tests indicated a significant predictive ability for the models for all species ($P < 0.001$) (Fig. S2). The percentages of the variable contribution by members of the *Chionaspis pinifoliae-Chionaspis heterophyllae* species complex to the model construction are shown in Table 1. To visualize the suitable habitat areas of the species complex, model predictions were imported into a geographic information system and the areas were reclassified into two arbitrary categories of habitat suitability: unsuitable habitat (<threshold) and habitat suitability areas (threshold-1). The threshold is shown in Table 1. The distribution of the cryptic species is determined by different responses to the environment (Figs. 2 and 3).

Suitable habitat areas for S1 were found at the high mean temperature of the driest quarter (Bio9) ($-1-21$ °C) and precipitation of the coldest quarter (Bio19) (200–5,000 mm). Both climate variables are the most important predictors for their potential distribution (Fig. 2 and Table 2). The mean temperature of the driest quarter (Bio9) was also an important environmental factor constraining the distribution of S2, with the highest suitability at values between 14.5 °C and 21 °C. The distribution of S5 was mainly constrained by the mean diurnal range (Bio2) (14.5–20.5 °C). S6 showed optimum suitability at the mean temperature of the wettest quarter (Bio8) ($-5-5$ °C) and at the mean temperature of the driest quarter (Bio9) ($-3-31$ °C). S7 was mainly constrained by the precipitation of the coldest quarter (Bio19) (100–900 mm). The mean temperature of wettest quarter (Bio8)

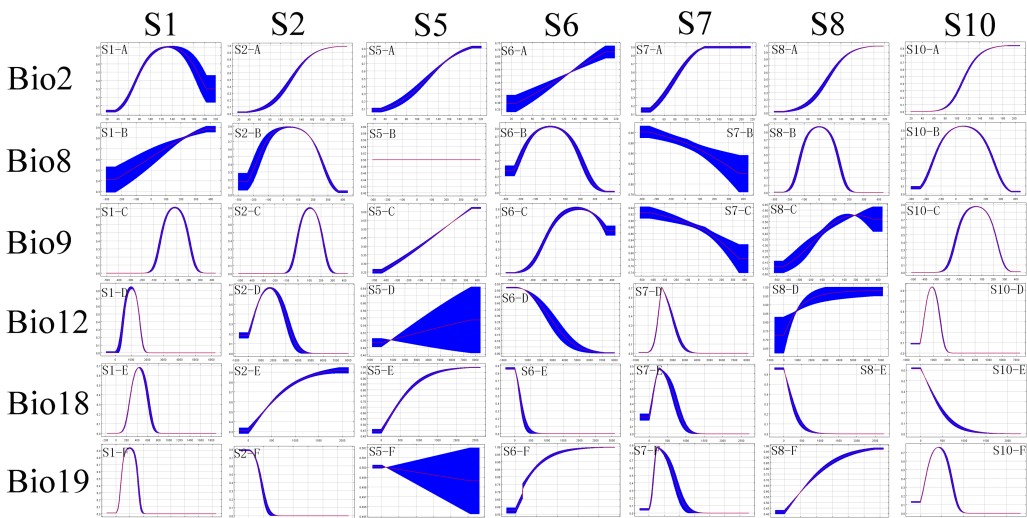

**Figure 2** **The ecological response curves for each cryptic species.** The response curves are based on the ENMs. Response curves show the ranges in environmental conditions that are more favourable for the distribution of the species. The x-axis of the variables represents their ranges for the complete study area, while the y-axis represents the predicted suitability of focus species.

was an important climatic factor constraining the distribution of S8, with the highest suitability at values between −2 °C and 12 °C. Suitable habitat areas for S10 were found for the mean diurnal range (Bio2) of 2 °C–15 °C, and the mean temperature of the wettest quarter (Bio8) of −20 °C–30 °C.

Relative to other environmental variables, temperature factors including mean diurnal range (Bio2) and mean temperature of the wettest quarter (Bio8) were the main factors affecting the distribution of this species complex.

## Niche overlap, niche equivalency and similarity

The results from the niche overlap suggest high variations exist in the environmental space inhabited by the different CPCH (Table 3). The PCA-env of niche similarity (Figs. S3–S6) in the species pairs of two PCA axes explaining from 58.7% (S1 vs S5) to 77.48% (S6 vs S10) of the total variation of the seven climatic variables. A great overlap was shown for certain pairs of species, such as: S6 and S8 (0.649), S2 and S5 (0.488), S2 and S7 (0.477) and S7 and S8 (0.477). However, the niche overlap of the other species pairs was relatively low, such as S1 and S6 (0.014), S1 and S7 (0.042), S2 and S8 (0.0146), and S2 and S6 (0.097). The low niche overlap suggested that they occupy considerably different environmental niches. All niche overlap values are illustrated in Table 3.

According to the identify test, the true calculated niche overlap of all species pairs, are outside of the 95% confidence interval of the null hypothesis (Fig. S7) and confirm the separation between them. Thus, these species pairs showed that the species' ENMs were not equivalent.

The above test results indicate that the niches of CPCH species complex are similar, but they are not identical.

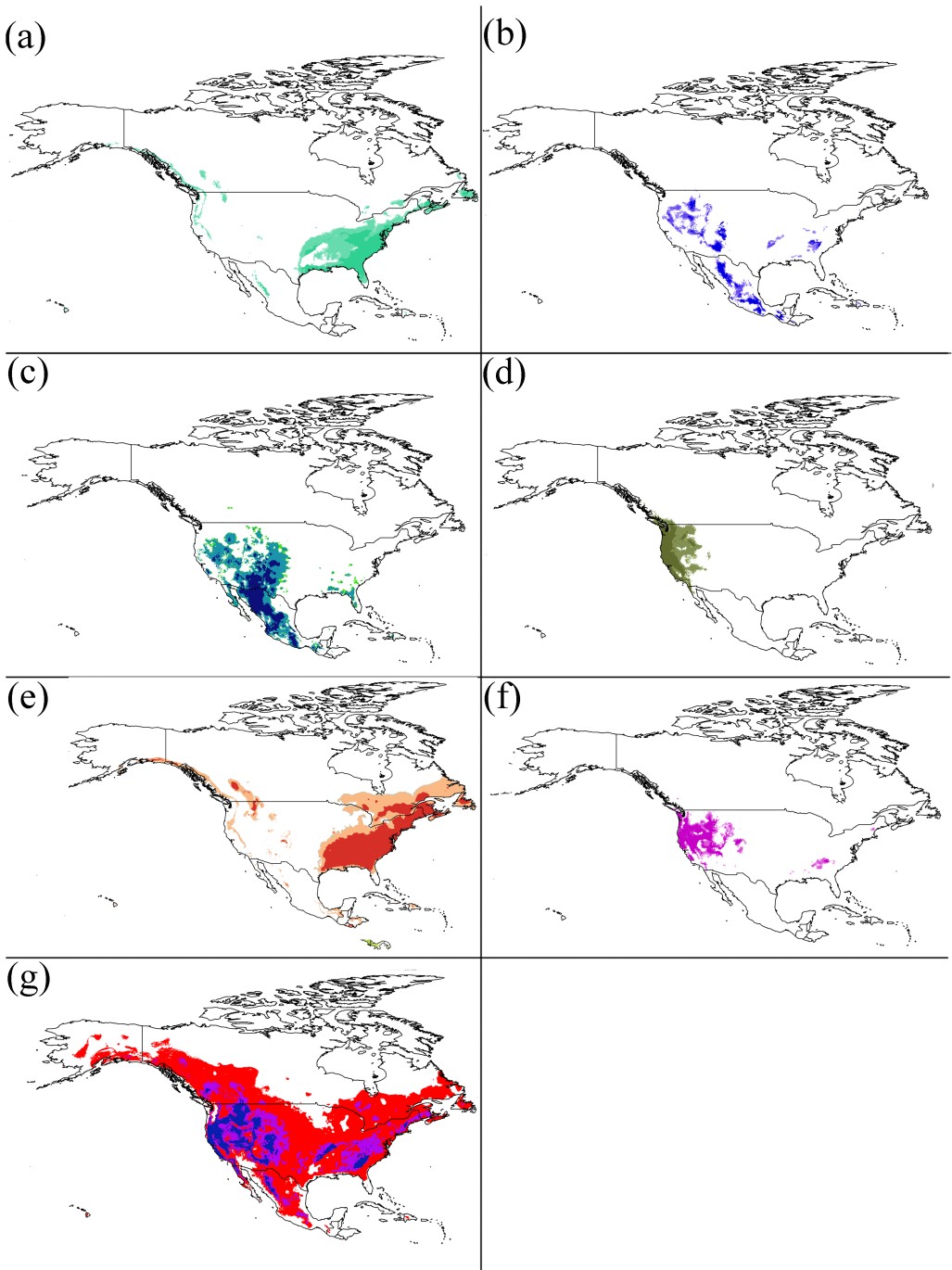

**Figure 3 Potential distributions of each cryptic species.** For (A–G) represents the areas that have high suitability after applying the LPT (minimum training presence threshold). (A) S1, (B) S2, (C) S5, (D) S6, (E) S7, (F) S8, (G) S10. Niche models results were modelfied in ArcGis 10.1 (Environmental Systems Research Institute, Redlands, CA, USA). The base map was created with Natural Earth Dataset (http://www.naturalearthdata.com).

**Table 2  Percentage of variable contribution to the model construction, derived from the permutation importance analysis from MaxEnt.** For each armored scale insect taxon, the two variables with highest contributions are presented in bold.

| Variable | S1 | S2 | S5 | S6 | S7 | S8 | S10 |
|---|---|---|---|---|---|---|---|
| Mean diurnal range (Bio2) | 6.3 | **33.7** | **65.8** | 1 | **17.4** | **35.2** | **36.7** |
| Mean temperature of wettest quarter (Bio8) | 1.8 | 7.9 | 0.2 | **43.3** | 2.8 | **47.3** | 23 |
| Mean temperature of driest quarter (Bio9) | **30.6** | **33.6** | 3.5 | **34** | 4.6 | 5.8 | 17.6 |
| Annual precipitation (Bio12) | 20.3 | 5.9 | 0 | 0 | 1.5 | 0.3 | 2.1 |
| Precipitation of warmest quarter (Bio18) | 8 | 14.5 | **30.5** | 15.8 | 15.8 | 0.1 | 0.1 |
| Precipitation of coldest quarter (Bio19) | **33** | 4.3 | 0 | 6 | **58** | 11.2 | 20.4 |

**Table 3  Niche comparisons for the cryptic species complex.** Niche overlap values are presented for the comparisons of niche similarity and equivalency of species 1 with species 2. All of the comparisons between the species highlight the nonequivalency of their niche.

| Species | | Niche overlap (D) | Niche similarity | |
|---|---|---|---|---|
| 1 | 2 | | $1 \rightarrow 2$ | $1 \leftarrow 2$ |
| S1 | S2 | 0.36 | Similar* | Similar* |
| | S5 | 0.476 | Similar* | Similar* |
| | S6 | 0.014 | ns | ns |
| | S7 | 0.0421 | Similar* | Similar* |
| | S8 | 0.142 | ns | ns |
| | S10 | 0.231 | Similar* | ns |
| S2 | S5 | 0.488 | Similar* | Similar* |
| | S6 | 0.097 | ns | ns |
| | S7 | 0.477 | Similar* | Similar* |
| | S8 | 0.0146 | ns | ns |
| | S10 | 0.351 | Similar* | Similar* |
| S5 | S6 | 0.158 | ns | Similar* |
| | S7 | 0.39 | ns | Similar* |
| | S8 | 0.345 | Similar* | Similar* |
| | S10 | 0.325 | Similar* | Similar* |
| S6 | S7 | 0.246 | Similar* | Similar* |
| | S8 | 0.649 | Similar* | Similar* |
| | S10 | 0.37 | Similar* | Similar* |
| S7 | S8 | 0.477 | Similar* | Similar* |
| | S10 | 0.437 | Similar* | Similar* |
| S8 | S10 | 0.325 | ns | Similar* |

**Notes.**
ns,  not significantly different.

## Geography of speciation

The degree of geographic overlap by ENM between the species pairs provides evidence that suggests that some sympatric, parapatric and allopatric species exist in our analysis based on the similarity test. Of the 21 total pairs, two cases of parapatry between species pairs in these species complex were known (S1 and S7; and S1 and S10). Four cases of allopatry

between species pairs in these species complex were known (S1 and S2; S1 and S5; S1 and S6; and S1 and S8). In addition, 15 species pairs in the species complex were found to exhibit sympatry. The parapatric species pairs had a small niche overlap (S1 vs S7 = 0.041; S1 vs S10 = 0.231). As predicted, all allopatric (except S1 vs S5 = 0.476) species pairs also suggest a small niche overlap. All speciation types also showed niche divergence in same direction.

## DISCUSSION

In this study, the environmental constraints for the current distribution of the members of the CPCH species complex were identified by applying ecological niche modeling and ordination techniques. The distribution range of the CPCH species complex in the North American continent mirrors the various environmental conditions to which they are adapted to. Additionally, they may also reflect the physiological differences between them. Temperature factors were the most significant limiting factor in the distribution of all CPCH. Higher temperatures could increase the fitness and abundance of the scale insect (*Dale & Frank, 2017*), thereby possibly increasing their chances of survival in extreme conditions.

Our results have shown that the distribution of each member of the CPCH complex is constrained by a set of environmental conditions. Climate change will directly affect the potential distribution of species, the ranges of which are strongly constrained by temperature and precipitation (*Chen et al., 2011*). The potential distribution of the members of the CPCH complex under climate change is not only limited by climate variables but also affected by other factors, such as species dispersal mechanisms, host-plant availability and human-mediated transport. The results provide a distribution range to facilitate the further control of these insects and formulate quarantine measures when an invasion by a member of this complex occurs.

Given the wide variation in the environmental conditions at locations where the members of the CPCH complex can be found, a low niche overlap between some members of this complex was expected. The low values of niche overlap between S1 and S7, S1 and S6, S2 and S6, and between S2 and S8, were also reflected in their different climate variable constraints. The result of the niche equivalency test between all pairs suggested a lack of ecological exchangeability. This result is in agreement with a previous study (*Scriven et al., 2016*) that morphological similarity does not necessarily equate to ecological equivalence. The result of the niche similarity test shows that the CPCH complex shares more climate niche characteristics than would be expected from random occurrence. In summary, the results show that there is a close relationship between these species, and they are likely to share climatic niche spaces. It is further confirmed that the members of the CPCH complex are closely related but are of different taxa. The differences in the environmental constraints of the different CPCH complex are also reflected in the niche similarity, overlap and equivalency results. In our analysis, the significant differences in the niche spaces reflect the reported taxonomic divisions within the CPCH complex (*Gwiazdowski et al., 2011*). Previous research has used differences in niches to support species delimitation

(*Aguirre-Gutiérrez et al., 2013*; *Raxworthy et al., 2007*). Our results support the current species delimitation of the CPCH complex by molecular methods (*Gwiazdowski et al., 2011*; *Vea, Gwiazdowski & Normark, 2012*). These cryptic species are morphologically similar, but the climatic ecological characteristics have caused differentiation. Our results are consistent with other research that niches are similar but not equivalent across closely related species (*Aguirre-Gutiérrez et al., 2013*).

There are three traditional geographic categories for the phenomenon of speciation, which are allopatric (non-overlapping), sympatric (overlapping) and parapatric (adjoining) speciation, which depend on the degree of range overlap between species pairs during the speciation process (*Zheng et al., 2017*; *Bultin, Galindo & Grahame, 2008*). The speciation of different geographic categories was identified by the potential distribution modelling in recent studies. Our study supports the hypothesis that the niche overlap in the sympatric population was greater than that of the allotropic population (*Hochkirch & Gröning, 2012*).

Considerable overlap was found in the climatic niches of the different members of the CPCH complex (Table 3), such as S6 vs S8, S1 vs S5 and so on. These findings suggest that the divergent natural selection along climatic axes of the niche had played a limited role in the development of some pairs of this species complex. Many studies on a wide variety of taxa have suggested that ecological divergence plays an important role in sympatric populations (*Via, 2001*; *Schliewen et al., 2001*). Our results in the current paper suggest the lack of a large proportion of niche divergence between species pairs. These results are in accordance with the expectation that, given the climatic conditions available to them, closely related pairs will not be equivalent in their climatic niches, but will typically be more similar than expected. Our results agree with previous studies on the niche conservatism of sister species or species complexes (*Graham et al., 2004*; *Silva et al., 2014*). In our study, sympatric species pairs had very similar environmental niches or relatively high niche overlap, e.g., S6 vs S8 (Table 3, Fig. 2). This result suggests that these cryptic species likely experienced similar environmental pressures throughout their evolutionary history. Furthermore, the climatic niches are less likely to be influenced by competition and may thus represent interspecific differences in the fundamental niche. However, the larger niche overlap must lead to competition in sympatric and parapatric species pairs. Direct competition between these cryptic species is unlikely to exist (e.g., feeding), but there is potential competition for resources (e.g., host plants). The differences found in their use of forage plants may possibly be driven by competition and could reflect differences in their realized niches, rather than their fundamental niches.

Allopatric speciation is essentially a spatial process where two populations become genetically isolated due to geography, and thus the species pairs would have a relatively small niche overlap. The results of the current study suggest that some allopatric species pairs had a relatively small niche overlap, such as: S1 vs S6 and S1 and S8 (Table 3, Fig. 2). These results indicate that their niches belong to different environmental conditions. A hypothesis has been proposed that allopatric species may originate when a geographic barrier (i.e., an area of unsuitable environmental conditions between two sets of populations) develops faster than adaptation to these new ecological conditions. S1 is mainly distributed in southeastern North America and S6 located in western North America. There is a relatively

large distance between the two species, with significant geographical isolation. Similarly, there is a larger geographic barrier between S1 and S8. In addition, the host plants of S1 are *Pinus sylvestris*, *P. nigra*, *P. palustris*, *P. echinata*, *P. pungens*, *P. taeda*, *P. virginana*, *P. elliottiii*, and *P. rigida*, while the host plants of S6 are *P. torreyana*, *P. strobiformis*, *P. lambertiana*, *P. ponderosa*, *P. contorta*, *P. quadrifolia*, *P. attenuate*, *P. halepensis*, *P. jeffreyi*, *P. sabineana*, and *P. torreyana var. insularis*. The host plants of S8 are *Pinus undet*, *P. radiate*, *P. lambertiana*, *P. attenuate*, *P. ponderosa*, *P. contorta* and *Pseudotsuga menziesii*. The species pairs have completely different host plants. Therefore, there may be a strict dietary isolation between the two species pairs. The distribution of host plants might determine the distribution of insects in our studies (*Provencher et al., 2005*). However, since populations on each side of the barrier would still have the same niche in cryptic species, it would have been niche conservatism that maintained the allopatric distribution. In our study, a small but clear overlap between the suitable areas defined by the ENMs of the allopatric sister taxa (*Kozak & Wiens, 2006*) would have been found. Our study also supports the hypothesis that niche conservatism may be generally important in allopatric speciation because it will limit adaptation to ecological conditions at the geographic barrier (*Warren, Glor & Turelli, 2008*). Parapatric species pairs (e.g., S2 vs S8, S5 vs S6, S2 vs S6) also show a relatively low niche overlap value. The underlying mechanism is similar to that of allopatric species distribution.

## CONCLUSION

In summary, our results suggest that niche conservatism is common in this cryptic species complex and that these species occupy different climatic niches. The results suggest that allopatric species pairs had a relatively small niche overlap, which indicates that their niches are adapted to different environmental conditions. In addition, the results in our current paper suggest that there is a lack of a large proportion of niche divergence between sympatric species pairs. The results suggest that these cryptic species had likely experienced similar environmental pressures throughout their evolutionary history. In current study, the niche comparisons were implemented only using the *Broennimann et al. (2012)* method. However, other approaches, such as ENMtools (*Warren, Glor & Turelli, 2010*) or using more sophisticated methods of multivariate analysis, such as measures of the multidimensional overlap in species' niche positions and breadths (*Blonder et al., 2014*) might provide much clearer results. Additional, only climate condition was consider in current work, however, other niche factor such as dispersal ability (*Guisan & Thuiller, 2005*), Interspecific interactions (*Gao & Reitz, 2017*) and host-plant availability (*Ning, Wei & Feng, 2017*) also affected the precision of the model. Thus, these factors should be consideration in future research.

### Funding

This work was supported by the National Science Foundation Project of China (no. 31301899, no. 31501876 and no. 31872272), the Natural Science Foundation of Shanxi

(no. 201601D021122) and Shanxi Agricultural University of Science and Technology Innovation fund projects (2015YJ03). The funders had no role in study design, data collection and analysis, decision to publish, or preparation of the manuscript.

## Grant Disclosures
The following grant information was disclosed by the authors:
National Science Foundation Project of China: 31301899, 31501876, 31872272.
Natural Science Foundation of Shanxi: 201601D021122.
Shanxi Agricultural University of Science and Technology Innovation fund projects: 2015YJ03.

## Competing Interests
The authors declare there are no competing interests.

## Author Contributions
- Qing Zhao performed the experiments, analyzed the data, prepared figures and/or tables, approved the final draft.
- Hufang Zhang conceived and designed the experiments, contributed reagents/materials/analysis tools, authored or reviewed drafts of the paper, approved the final draft.
- Jiufeng Wei conceived and designed the experiments, performed the experiments, analyzed the data, contributed reagents/materials/analysis tools, prepared figures and/or tables, authored or reviewed drafts of the paper, approved the final draft.

## Data Availability
The raw measurements are available in Files S1 and S2.

## Supplemental Information
Supplemental information for this article can be found online at http://dx.doi.org/10.7717/peerj.7042#supplemental-information.

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
