# Peer review of "Climatic niche comparison across a cryptic species complex"

_PeerJ, doi:10.7717/peerj.7042_

## Round 0.1 · original submission · Major Revisions

The reviewers valued your work and had constructive comments to improve it. The two tests that are applied needed to explained in more detail and reconsidered in the light of the reviewers remarks. The selection/regularization needs to be made more strict using the more recent insights presented by the reviewers, including the usage of AUC vs Omission rates. The remarks on circularity of reasoning needs to be addressed as well. From my own reading I was surprised to see that the detailed Maxent modelling did not have a follow-up in the later testing approaches that used blunt PCA axes. I found this a kind of inconsistency in the paper that needs to be discussed. One could measure the niche similarity also by looking at how well a Maxent model for one species fits that other.

Details: L 176 the threshold is undefined.

L 325-329: If you feel this way, please apply the more sophisticated methods; alternatively argue why you do not. Note that the first reviewer says that the methods are NOT state-of-the-art anymore...

Reviewer 1 ·

Basic reporting

This manuscript is very clear and mostly well structured, necessary data is publicly available. However, some methods are included in the results (line 175-178). I suggest some further literature review in the niche modeling area especially regarding methods and the history of the field which they keep presenting as new although it has a history of more than 20 years. Results are relevant although some methodological concern will be presented later.

Some small typos have to be fixed including the confusion between allopatry and allotropic in the abstract. A small wording concern regarding the use of the word niche to refer to the climatic niche should be assessed. This becomes particularly relevant when reaching the discussion/conclusions where other niche axes are discussed such as host plants but are to considered part of the niche.

Experimental design

The research is within the scope of the journal. The research question is well defined and relevant to several biology fields including biogeography and climate change research. Methods are replicable although not state of the art as stated by the authors in line 231. Also in the methods description line 111 authors state ENM's "simulate suitable environments" I urge them to re-write this as ENM constructions does not involve any simulation of data but rather construction of models that are interpolated and extrapolated in space.

I have some detailed concerns below:

1- Maxent is known to produce overly complex models that contain more parameters than observations. This is particularly true in cases of low sample sizes like this one and directly affects the estimation of species distributions and in this case all the conclusions regarding climatic niches. For example, with overfit models it is to be expected that niches of geographically separate species will be different as they are geographically overfit. This problem is easily to overcome by not using defaults in the modeling process but rather comparing different combinations of feature classes and regularization multipliers. This can of course be done manually but also using freely available software. Here are two possibilities the authors should read and consider:
a- Brown JL (2014) SDMtoolbox: a python-based GIS toolkit for landscape genetic, biogeographic and species distribution model analyses. Methods in Ecology and Evolution, 5, 694–700.
b- Muscarella R, Galante PJ, Soley-Guardia M et al. (2014) ENMeval: An R package for conducting spatially independent evaluations and estimating optimal model complexity for Maxent ecological niche models. Methods in Ecology and Evolution, 5, 1198–1205.

2- Why choose a 5km grid for small insects? Is there some estimate of their dispersal capabilities that lead to this choice? It appears to me that at a 5km grid scale (about 25 square km) it is to be expected that sympatric species are experiencing the same average climate but probably not at a more microclimatic scale.

3- Several papers have stated problems with using AUC as a metric of model performance and suggested many alternatives including Omission rates and partial ROCs. However, the authors chose to ignore these criticisms and use AUC with no mention of the problems. I believe they have to use a different measure or at least mention the potential shortcomings and why they think they are not important. I am suggesting some relevant literature but this is by no means comprehensive:
a.Peterson AT, Papeş M, Soberón J (2008) Rethinking receiver operating characteristic analysis applications in ecological niche modeling. Ecological Modelling, 213, 63–72.
b. Lobo JM, Jiménez-valverde A, Real R (2008) AUC : a misleading measure of the performance of predictive distribution models. , 145–151.

4- Why using the "maximum training sensitivity combined with specificity" this is not justified here. Conclusions of this paper are heavily dependent on this choice since it determines the area for which the environmental variables are extracted to compare the climatic niches. Probably it would be better to try several thresholds that are used in the literature and see how robust the results are. Especially it strikes me as odd that they are not using the very widely accepted T10 and LPT thresholds. For example see:
Pearson RG, Raxworthy CJ, Nakamura M, Peterson AT (2007) Predicting species distributions from small numbers of occurrence records: a test case using cryptic geckos in Madagascar. Journal of Biogeography, 34, 102–117.

5- Related to point 4. Lines 175-178 they authors chose an arbitrary division of the suitability for their comparisons. This is wrong. The interpretation of suitability is species dependent and what is suitable for one is not necessarily suitable for the other. This would be easier to understand when using LPT as it shows how the lowest presence threshold can vary widely between models from things as low as 0.06 to anything and so 0.4 can be high suitability for certain species.

6- Why test all possible pairs for niche equivalency? The niche conservatism hypothesis refers to sister species and so that conclusion will only make sense when testing sisters and having them be equivalent/divergent.

Validity of the findings

There is a lot of speculation not stated as so. Specially when dealing with how these finding support the molecular division of the species. These conclusions are very circular and should be acknowledged as so. Indeed, models were built based on the molecular subdivision and so the fact that the models differ only allows for speculation of how this supports the molecular hypothesis. If the authors want a real test they would have to creat a "null" model in which the combined points for the complex are divided in different species and modeled and those models compared. If observed results are different than the randomization the they could draw conclusions on the support for this taxonomic hypothesis.

Additional comments

This paper has very interesting questions and the conclusions could be relevant to different fields. However, current methodology brings the results into question and do not allow for robust answers to the questions. I believe methodological fixes are easy to implement and conclusions will likely change and bring more light into the history of this very interesting species complex.

·

Basic reporting

Overall the text is clean, easy to read, and unambiguous

Experimental design

The research questions are well defined and appropriately test. One exception regards the parameterization of the ENMs. It much better use omission error rate vs. AUC for model parameterization. AUC alone tends to result in overfit models (see Radosavljevic & Anderson,2014). For example, looking at your response curves for s10, they clearly overfit
as presented and do not have any biological realism. Further, in the future, its best to use spatial jackknifing, tests of multiple regularization parameters, and independent tests of feature classes for model parameterization. These factors are particularly useful for reducing model overfitting. Spatial jackknifing has demonstrated clear advantages over a random sampling of test/training occurrence data. In experiments on the effects of these two treatments, randomly partitioned occurrence datasets produced inflated estimates of performance and led to over-fit models (Radosavljevic & Anderson,2014).

For future research please consider using SDMtoolbox (my own software, cited below) or ENMeval for evaluating these factors.

See: Brown JL, Bennett JR, French CM (2017). SDMtoolbox 2.0: the next generation Python-based GIS toolkit for landscape genetic, biogeographic and species distribution model analyses. PeerJ


Radosavljevic, Aleksandar, and Robert P. Anderson. "Making better Maxent models of species distributions: complexity, overfitting and evaluation." Journal of biogeography 41.4 (2014): 629-643.

Validity of the findings

My main critique regards the author's interpretations of their niche equivalence and similarity tests. In 2008, Warren et al. (2008) proposed a pair quantitative tests of niche similarity: 1. an Identity Test that tested if two niches are equivalent (based on correlative distribution models) and, 2. a Background Test, that tested when Identity tests were significant, whether they are simply more similar than expected by chance. The Background test was aimed to test the power of the Identity Test, asking the question if the two distribution models are equivalent due to matching environments available in the habitat. If habitats contain identical environments, then species’ niches might be statistically equivalent solely due to the lack of different environments between distributions of the species. In 2012, Broennimann et al. introduced two complementary tests in Environmental space (vs. Geographic space of Warren et al. 2008, 2010). Though the two tests were conceptually similar, they renamed their tests to Equivalence and Similarity tests, corresponding to the Identity and Background Tests of Warren et al. 2008, respectively. However, in the manuscript Broennimann et al. they did a poor job explaining the exact purpose of each statistical test, so much so, the similarity test is widely misused or misinterpreted. Like Warren’s Identity tests, the Similarity test of Broennimann should only be used to detect the power of the Equivalence test to detect a difference. Thus, it only provides context to a non-significant Equivalence test and it should not be a separate test to detect ‘similarity’ among species. Please re-read Warren et al 2008 and in that context reread Broennimann et al 2012.

*Please change your citation from Broennimann et al. 2016 to Di Cola et al 2017

Under my interpretation, their results provide strong evidence of niche divergence (as all equivalence tests are all significant which equals non-equivalent niches), not niche conservatism. Further, given all significant equivalence tests, there is no need to interpret similarity test results.

One important critique that needs to be clarified. In the supplementary plots of niche overlap - all equivalence test show that only 1 iteration has been performed. Is this an error? Please confirm that you did perform 100 iterations of each equivalence test.

Warren, D.L., Glor, R.E. & Turelli, M. (2008) Environmental niche equivalency versus conservatism: quantitative approaches to niche evolution. Evolution, 62, 2868-2883.
Warren, D.L., Glor, R.E. & Turelli, M. (2010) ENMTools: a toolbox for comparative studies of environmental niche models. Ecography, 33, 607-611.
Di Cola, V., Broennimann, O., Petitpierre, B., Breiner, F.T., D'amen, M., Randin, C., Engler, R., Pottier, J., Pio, D. & Dubuis, A. (2017) ecospat: an R package to support spatial analyses and modeling of species niches and distributions. Ecography, 40, 774-787.
Broennimann, O., Fitzpatrick, M.C., Pearman, P.B., Petitpierre, B., Pellissier, L., Yoccoz, N.G., Thuiller, W., Fortin, M.J., Randin, C. & Zimmermann, N.E. (2012) Measuring ecological niche overlap from occurrence and spatial environmental data. Global ecology and biogeography, 21, 481-497.

Additional comments

Great job and cool study. I look forward to your future research. Do remember to properly tune your models in the future.

---

## Round 0.2 · Minor Revisions

Dear Jiufeng Wei,

Thank for the improved version of your paper. There are a few issues remaining. Please address the issues raised by the reviewer. In addition, I felt you did not address the issue I raised on the first version:

"From my own reading I was surprised to see that the detailed Maxent modelling did not have a follow-up in the later testing approaches that used blunt PCA axes. I found this a kind of inconsistency in the paper that needs to be discussed. One could measure the niche similarity also by looking at how well a Maxent model for one species fits that other. " Please do so in the revision.

I look forward to the revision. Yours sincerely and best wishes for 2019

Cajo ter Braak

Reviewer 1 ·

Basic reporting

Overall the text needs some english revision mainly in the newly added parts.

Experimental design

I thank the authors for thoroughly addressing my methodological concerns.

Validity of the findings

I thank the authors for thoroughly addressing my previous concerns with this manuscript I believe the new analyses help make this manuscript stronger. One of my comments remains to be addressed, as stated in my previous review:
"6- Why test all possible pairs for niche equivalency? The niche conservatism hypothesis refers to sister species and so that conclusion will only make sense when testing sisters and having them be equivalent/divergent."
In this same direction I believe their new results give support to niche divergence and not niche conservatism as stated in the abstract section.

---

## Round 0.3 · Minor Revisions

Please delete the extra redundant space in the abstract (first line). The rebuttal to the previous remarks was kind of evasive, which I regret but reluctantly accept. Nevertheless, please consider whether you dealt with in the abstract sufficiently with the remark by the reviewer "In this same direction I believe their new results give support to niche divergence and not niche conservatism as stated in the abstract section."

---

## Round 0.4 · Minor Revisions

The abstract can be improved in my view. My suggestion:
Our results also showed that the niche overlap had high variations in a climatic space inhabited by different cryptic species and niche divergence was support in same direction. The results also support the current species delimitation of the Chionaspis pinifoliae heterophyllae complex by molecular information, meanwhile, our study also supports the hypothesis that the niche overlap in the sympatric population is higher than that of the allopatry population. Moreover, the results provide...

->
Our results showed high variability in niche overlap, and more often niche conservatism than niche divergence. The results support the current species delimitation of the Chionaspis pinifoliae heterophyllae complex by molecular information. Our study supports the hypothesis that the niche overlap in the sympatric population is higher than that of the allopatry population. Moreover, the results provide....

Also, you did not answer to my question already asked in the first round and repeated in the revision: "From my own reading I was surprised to see that the detailed Maxent modelling did not have a follow-up in the later testing approaches that used blunt PCA axes. I found this a kind of inconsistency in the paper that needs to be discussed. One could measure the niche similarity also by looking at how well a Maxent model for one species fits that other. " Please discuss this point in the rebuttal.

---

## Round 0.5 · Minor Revisions

Unfortunately I cannot accept the current version due to the following issues:

L113 Did you do the test on the small subset or did Yackulicet al do it? If the first is true, why cite Yackulic? I could not find it so quickly in that paper... but I might have missed it. The paper could have been cited in a more general setting describing your data.

L152 It is not absolutely clear from the text whether you used PCA on all variables (1) or PCA on the variables selected to reduce to pairwise correlations (2). Part of the reason is that you use before the “Pearson correlation test” the word dimension reduction which is usually done using PCA. You reduce twice in case 2.

L240 I do not understand why you say “spatial autocorrelation”. In the text, on line 97, you just mention covariance between environmental variables.

L243 Say first why it is an inconsistency. The fact that two other papers use the same approach does not make you immune to the criticism. It says perhaps that your approach is not very novel.

L243 “Actually, many researches show that the Maxent model for one species fits that other also can be measure the niche similarity, such as ENMtools (Suarez-Mota et al., 2015).” I cannot understand this sentence. Please revise.

Please have your final version corrected and have your English corrected, with particular attention to the changes you make. I will accept or reject the next version.

---

## Round 0.6 · accepted · Accept

Thank you for checking the comments and the text. The article is Accepted.